# Urinary Tract Tumor Organoids Reveal Eminent Differences in Drug Sensitivities When Compared to 2-Dimensional Culture Systems

**DOI:** 10.3390/ijms23116305

**Published:** 2022-06-04

**Authors:** Yi Wei, Bastian Amend, Tilman Todenhöfer, Nizar Lipke, Wilhelm K. Aicher, Falko Fend, Arnulf Stenzl, Niklas Harland

**Affiliations:** 1Center for Medicine Research, Eberhard Karls University, 72072 Tuebingen, Germany; weiyimed@gmail.com (Y.W.); niz.lipke@gmail.com (N.L.); aicher@uni-tuebingen.de (W.K.A.); 2Department of Urology, University Hospital, 72076 Tuebingen, Germany; bastian.amend@med.uni-tuebingen.de (B.A.); todenhoefer@studienurologie.de (T.T.); urologie@med.uni-tuebingen.de (A.S.); 3Institute for Pathology, Eberhard Karls University, 72076 Tuebingen, Germany; falko.fend@med.uni-tuebingen.de

**Keywords:** bladder cancer, drug screening, bladder cancer organoids, BH3 mimics

## Abstract

Generation of organoids from urinary tract tumor samples was pioneered a few years ago. We generated organoids from two upper tract urothelial carcinomas and from one bladder cancer sample, and confirmed the expression of cytokeratins as urothelial antigens, vimentin as a mesenchymal marker, and fibroblast growth factor receptor 3 by immunohistochemistry. We investigated the dose response curves of two novel components, venetoclax versus S63845, in comparison to the clinical standard cisplatin in organoids in comparison to the corresponding two-dimensional cultures. Normal urothelial cells and tumor lines RT4 and HT1197 served as controls. We report that upper tract urothelial carcinoma cells and bladder cancer cells in two-dimensional cultures yielded clearly different sensitivities towards venetoclax, S63845, and cisplatin. Two-dimensional cultures were more sensitive at low drug concentrations, while organoids yielded higher drug efficacies at higher doses. In some two-dimensional cell viability experiments, colorimetric assays yielded different IC_50_ toxicity levels when compared to chemiluminescence assays. Organoids exhibited distinct sensitivities towards cisplatin and to a somewhat lesser extent towards venetoclax or S63845, respectively, and significantly different sensitivities towards the three drugs investigated when compared to the corresponding two-dimensional cultures. We conclude that organoids maintained inter-individual sensitivities towards venetoclax, S63845, and cisplatin. The preclinical models and test systems employed may bias the results of cytotoxicity studies.

## 1. Introduction

Urothelial carcinomas (UCs) are among the most frequent malignancies recorded in the urinary system [1,2]. Based on the anatomical situation where a carcinoma developed, physicians discriminate between upper tract urothelial cell carcinoma (UTUC) and bladder cancer (BC). UTUCs derive from the pyelocaliceal cavities and ureter, while BC develops in the bladder and urethra. UTUCs are not frequent, and the incidence rate ranges between 5 to 10% of all UCs diagnosed [1]. However, about 20% of patients diagnosed with UTUC develop BC eventually [3,4]. Based on pathological analyses, BC is discriminated in different stages reflecting the tumor size, invasion in muscle tissue of the bladder, involvement of lymph nodes, and generation of metastases [2,5]. Most UCs are initially superficial, but approximately 20% of patients diagnosed with carcinoma in situ develop muscle invasive BC [5,6].

Therapy of UTUC depends on the clinical stage of the malignancy, tumor grade, and the individual health risks of the patient [4,7,8,9]. For low-risk tumors, such as locally confined UTUCs without metastases, kidney-sparing, if possible, minimally invasive surgery is the preferred regimen [8]. Depending on the clinical situation of a patient, endoscopic ablation or ureteral resection can be considered [10,11]. For high-risk UTUCs with metastases, radical nephroureterectomy (RNU) is the standard of care [12,13,14]. A recent study indicated that RNU yields the best benefit with metastases in only one location [15], while UTUC patients with a more complex metastatic situation had better prognosis with chemotherapy [16]. For treatment of BC, the transurethral resection of the bladder tumor (TURBT) in combination with adjuvant therapy was recommended two decades ago [17]. Several recent studies are in line with this regimen. They recommend TURBT in combination with adjuvant intravesical chemotherapy or immunotherapy, while the role of neoadjuvant chemotherapy is currently under investigation [6,18,19,20,21,22]. For patients with muscle invasive bladder cancer (MIBC), cystectomy is a possibility [23,24], and a neoadjuvant combination therapy containing cisplatin (CIS) prior to cystectomy increased the overall long-term survival from 30% to 36% (*p* < 0.05) [25]. This is a significant increase in mathematical terms but not a substantial break-through for individuals affected. Currently, alternative regimens, e.g., immuno-oncology therapy or enfortumab vedotin, an antibody conjugate containing an antibody directed against nectin-4 and monomethyl auristatin E, is used in ongoing clinical trials [26,27]. The growing choice of treatment regimen will lead to an increasing importance of markers for individual treatment selection.

Further challenges in BC management remain. In some patients, the tumor fails to respond sufficiently to neoadjuvant or adjuvant therapy, while other patients develop resistance during the course of treatment [28]. Mechanisms contributing to drug resistance of cancer therapies include but are not limited to mutations of factors involved in regulation of cell viability and/or proliferation to transporter molecules secreting the active components out of the cytoplasm, non-coding RNAs, and the microenvironment of the carcinoma cells [29,30,31]. Long-term drug resistance was associated rather with slow proliferating cancer stem cells (CSC) [32,33,34,35].

Many novel BC therapies were developed by aid of tumor cell lines grown and tested in conventional two-dimensional (2D) cell culture vessels to explore various strategies for cytotoxic intervention and to manage cell proliferation, replicative senescence, apoptosis, necrosis, or mutagenesis [32]. Targeted therapies or molecular therapies were developed by interfering with distinct biochemical processes found predominantly or specifically in tumor cells. Such targeted or molecular therapies raised hopes for better BC management [36,37,38,39]. However, the complex interplay between proliferating tumor cells and neighboring cells [31], the influence of the extracellular matrix providing, e.g., anti-apoptotic signals by engaging integrins [40,41], the contribution of the vasculature, and other physiological aspects cannot be investigated by tumor lines or recombinant cells in meaningful ways in standard 2D systems only. Here, different animal tumor models came into play [42,43]. For some analyses, even humanized cancer models were developed for instance in immuno-deficient rodents [44]. However, pre-clinical tumor studies with animals raised a variety of concerns [45,46]. In addition, setting-up (patient-individual) cancer animal models is consuming a considerable amount of resources and frequently fails to recapitulate the etiology or pathology of cancer (of the individual patient) [47,48]. Three-dimensional (3D) in vitro tumor models tackling some of the above-mentioned disadvantages of 2D cell cultures could provide an additional platform for improved cancer research complementary to well-established technologies [49,50].

Organoids—as defined about a decade ago—are 3D in vitro cell culture constructs containing a scaffold providing a 3D mesh augmented by a blend of different cells [51,52]. Organoids may contain differentiation-competent stem cells and/or progenitor cells, which are capable of generating tissue-like structures mimicking the tissue of origin, at least in part [53]. In addition, organoids may contain epithelial or endothelial cells, either ex vivo or after in vitro differentiation of the progenitor cells, as well as mesenchymal cells [51,54]. In cancer research, tumor-derived organoids inherit the genome and mutations of the patient [55,56]. Embedding cells in organoids much better reflects the tissue situation of a tumor in situ [57,58]. In addition, setting up organoid cultures in multi-well plates may facilitate drug development in general as well as screening drugs with cells from an individual patient [59,60]. We therefore investigated the cytotoxicity of the DNA crosslinking drug CIS in comparison to two novel drugs, venetoclax (VTX) and S63845 (S63), interfering with regulation of the mitochondrial apoptosis pathway [61,62]. VTX (ABT-199) selectively targets BCL-2 and neutralizes its anti-apoptotic effects [63]. The S63 is a molecule attaching to the BH-3 binding site of the MCL-1 inhibitor with high affinity, thus facilitating apoptosis of cells [64]. We explored these three active components in standard 2D cultures versus 3D organoids [65,66]. To this end, we employed normal urothelial cells (NUCs), bladder cancer cell lines with known relative resistance to cisplatin in 2D culture (RT4: low grade and HT1197: high grade) [67,68], UTUCs (from tumor samples of patients #56, #147), and patient derived BCs (from tumor samples of patients #41, #44, #107, #136, #140). In 2D cultures, we compared the dose-responses and kinetics using a colorimetric WST assay versus a 2D chemiluminescence assay. In addition, we compared the cytotoxic effects of CIS, VTX, and S63 in 2D standard cultures versus 3D organoids using specific 2D- and 3D-chemiluminescence technologies, respectively.

## 2. Results

### 2.1. Upper Tract and Lower Tract Urinary Carcinoma Organoids

Organoid cultures were generated from two RNU and five TURBT surgery samples, respectively. In this study, cultures from two RNU samples, i.e., BCO#56 and BCO#147, and from one TURBT specimen, i.e., BCO#140, were included as they granted sufficiently long-term 3D growth as organoids in vitro and at the same time expansion of adherent cells in 2D cultures (Figure 1). Normal urothelial cells (NUCs) and the established BC cell lines HT1197 and RT4 served as controls (Figure 1) Significant differences in growth patterns and mitotic activity between the two UTUC organoids were not noted. The BC organoid BCO#147 tended to generate cystic organoids while BCO#56 and BCO#140 contained cells inside (Figure 1A–C). As observed in NUCs, HT1197, and RT4, adherent growth was noted when cells were extracted from organoids and seeded directly as 2D cultures in cell culture vessels (Figure 1 D–I). Figure 1 shows a representative experiment. In addition, the expression of cytokeratins (CKs) was investigated in bladder cancer organoids (BCOs) to determine the contribution of urothelial cells to the organoid cultures. Some cells expressed the CKs reactive with antibody AE1/AE3, while other cells failed to bind AE1/AE3 (Figure 2). CK5 and CK20 were detected in some but not all cells as well (Figure 2). Expression of vimentin characterized some cells in the organoids as mesenchymal cells (Figure 2) Fibroblast growth factor receptor 3 expression was recorded on virtually all cells (Figure 2). Figure 2 shows a representative experiment as well. The data confirmed that the organoids investigated contained urothelial as well as mesenchymal cells. Differences in staining patterns of the antigens investigated in this study between organoids from the BC sample in comparison to the RNU-derived organoids were not observed.

### 2.2. Comparing Different Viability Assays Using Urothelial Cells in Adherent Cultures

In a first series of experiments, drug responses to CIS, VTX, and S63, respectively, were explored by a chemiluminescence assay (CellTiterGlo 2.0; CTG) in direct comparison to a colorimetric assay using the water-soluble tetrazolium (WST) salt as the substrate and employing standard 2D cultures of adherent cells. To this end, NUCs and the BC lines HT1197 and RT4 were utilized. To avoid artifacts associated with cellular senescence of somatic NUCs from a single donor after extended in vitro passaging, we preferred to include early passage NUCs from three individual donors. Addition of CIS (Figure 3A,D), VTX (Figure 3B,E), or S63 (Figure 3C,F) yielded distinct dose-response curves by CTG assay after 2 days of incubation (Figure 3A–C) when compared to the WST assay (Figure 3D–F). By CTG assay, CIS was slightly more effective on NUCs when compared to the two BC lines (Figure 3A). In contrast, by WST assay, NUCs produced an artifact of a virtually high normalized viability index (NVI) in controls and at 0.5 μM CIS, but no significant difference in sensitivities at 1.0 μM CIS in comparison to HT1197 and RT4, respectively (Figure 3D). This artifact may be explained by proliferation of NUCs in the absence of CIS and under low CIS dosage over 3 days prior to the colorimetric assay. However, it was recorded only when using the WST chemistry and NUCs (Figure 3D). When using CTG assays, this effect was not observed (Figure 3A). We therefore considered this a technical artifact but not a result associated with CIS cytotoxicity. By CTG assay, NUCs were more sensitive to VTX-induced apoptosis when compared to HT1197 and RT4 cells (Figure 3B). In contrast, by WST analysis, this difference was not evident (Figure 3E). The cytotoxic effects of S63 yielded comparable results by CTG assay: NUCs were more sensitive to S63 when compared to the BC lines HT1197 and RT4 (Figure 3C). By WST analysis, the same trend was observed, but differences between NUCs and RT4 were less prominent (Figure 3F). Moreover, statistically significant differences were noted when CTG analyses of urothelial cells were compared to WST assays with the same cells (Figure 4). The artifact in the WST assays of NUCs after CIS treatment is evident here as well (Figure 4A). The NVI after CIS treatment was significantly different for the two tumor lines utilized, but the mean values remained in a comparable range (Figure 4A). In contrast, using VTX or S63 generated not only significant differences in NVI levels of all cells tested but in addition yielded eminent disparities of the means of the NVIs computed (Figure 4B,C). Table 1 summarizes these analyses. The data document showed that the colorimetric WST assay yielded quite different half maximal inhibitory concentrations (IC_50_) towards CIS, VTX, and S63 on NUCs, HT1197, and RT4 in standard 2D cultures after 1 to 3 days of incubation when compared to the CTG analyses employed here under otherwise identical conditions.

### 2.3. Individual Sensitivities to Cytotoxic Drugs in Two-Dimensional versus Three-Dimensional Cell Cultures

To investigate if cells in adherent 2D UC cell cultures yield distinct sensitivities to CIS, VTX, and S63 in viability assays in comparison to 3D organoid cultures, a chemiluminescence assay was employed. We assumed that absorption, ray diffraction, or reflections of the beam in an ELISA reader by organoids and Matrigel domes might bias the read-out. Therefore, analyses of cell viabilities were not performed in 3D organoids by colorimetric methods (e.g., WST assay in an ELISA reader). In contrast, to determine the cytotoxic effects and the IC_50_ of the components included in this study in 2D versus 3D cultures, we utilized a chemiluminescence technology (Figure 5). Adding different amounts of CIS to cells in 2D (Figure 5A) in comparison to the same cells in 3D organoids (Figure 5D) yielded overall comparable responses. Cells from patient #140 showed the highest sensitivity to this drug (Figure 5). A different response was observed upon addition of VTX. Cells from patient #147 yielded a high sensitivity towards the BH-3 mimic VTX in 2D (Figure 5B). However, in 3D organoids, these cells presented with the lowest sensitivity over the whole range of concentrations investigated (Figure 5E). This difference was also observed with S63 and cells from patient #147. The 2D cultures of cells from patient #147 responded well to S63 (Figure 4C), while in 3D organoids, BCO#147 was less sensitive (Figure 5F). In addition, statistically significant differences were recorded when the NVIs were computed after treatment of the cells with CIS, VTX, or S63 in 3D organoids vs. 2D cultures (Figure 6). A hypothesis that cells in 2D may be more sensitive to CIS when compared to the same cells in an organoid was not observed as BCO#56 yielded a significantly reduced NVI when compared to #56 cells in 2D. In contrast, for BCO#140 and BCO#147, the opposite was found (Figure 6A). Upon treatment with VTX, BCO#147 yielded a most prominent gap between the mean NVI when compared to the NVI of #147 cells in 2D, while the NVIs of BCO#56 and BCO#140 in comparison to the corresponding 2D cell were less, but still significantly, different (Figure 6B). Comparable results were obtained by S63 treatment of the same cells: A clear and significant difference in cell viabilities between BCO#147 when compared to #147 cells in 2D, and less prominent but significant differences between the NVIs of two other organoids and cells in 2D, respectively (Figure 6C).

The IC_50_ doses for CIS, VTX, and S63 of cells from patients #56, #140, and #147 in 2D adherent cultures in comparison to the IC50 of BCO#56, BCO#140, and BCO#147 were computed as well (Table 2). A trend indicating that cells in 2D cultures yielded in general a lower or higher IC_50_ when compared to the corresponding cells in organoids was not observed. Interestingly, cells from patient #140 generated the most harmonic results in 2D versus 3D experiments, while the cells from patients #56 and #147 generated distinct responses and differed between 3-fold to almost 10E05-fold in the cytotoxicity assays (Table 2). We conclude that the UC cells included in this study generated quite distinct viability or proliferation responses to CIS, VTX, and S63 in 2D versus 3D culture.

## 3. Discussion

Determining the viability of cells upon exposure to chemicals in different concentrations and over different periods of time plays a central role in pharmacology and toxicology research. Such basic experiments have been performed largely by using established cell lines [69,70,71]. Genetic studies contributing to a given disorder can be studied in cell lines in reproducible ways. However, research on BC cell lines inherits its disadvantages as well. Among them, a bias towards more proliferative cells selected in vitro during ongoing culturing is a factor [49]. Moreover, results from in vitro studies provided evidence that different tumor cell lines produced distinct sensitivity profiles and kinetics in cytotoxicity testing depending on the different technologies applied [72,73,74,75]. The assay technologies investigated represent a quite different chemistry and apparatus including reduction in tetrazolium salts (e.g., MTT, XTT, WST), detection of leakage of enzymes from dead cells (e.g., LDH release), determination of intracellular ATP levels (e.g., luciferase-based systems), labeling of DNA fragments (TUNEL-assays), and others. In other words, not only the individual cell or cell line under investigation during cytotoxicity testing but also the design of the assay has an influence on the results. This influence of the assay technology and apparatus on the outcome is relevant in clinical situations, when, for instance, cancer patients are either resistant to a standard component such as CIS or become resistant over the course of a therapy. In such cases, a standard regimen will follow a schedule of cancer therapies with different drugs based on clinical evidence [76]. Here, a personalized cancer therapy by rapid and robust testing of tumor cells of the individual affected to determine the most effective anti-cancer drug would save valuable time, especially for the patient. It may even increase the chances for long-term remission, if cancer stem cells which were associated with cancer recurrence are eliminated at least in part by the novel regime as well [30,33,34,77,78]. In a move towards individualized therapeutic regiments, biomarkers have an increasing role to determine which medical treatment might result in the best response [79,80]. Comparable approaches in UTUCs and BCs have resulted in the definition of different molecular subtypes of cancer cells [81] but fail to show a clear correlation to the response to medical treatment [82]. In this context, patient-derived organoids were discussed as an advanced cell culture system towards personalized cancer therapy [49,57,59,83,84,85].

We employed in our first study of effects of selected drugs on BCOs only the CTG method in the corresponding cytotoxicity assays. By this technology, we determined the overall effects of drugs to all cells in an organoid. Different methods may yield other results, including data closer to the patient’s in vivo sensitivity. In contrast to the 2D analyses, such a comparative study including several distinct methods to determine cell viabilities in BCOs must await future experiments. Of note, our very preliminary data suggest that other technologies may provide information on induction of cell death of individual cells in an organoid (data not shown). Such novel technology will eventually pave the way for targeting the tumor cells or even CSCs within a BCO with better efficacy and specificity, possibly even sparing stromal cells or normal urothelial cells. The clinical benefit of such research needs not to be emphasized here. However, this aspect is way beyond the focus of the current study.

Organoids come closer to the tumor situation in vivo when compared to standard tumor cell cultures. Organoids are built by a blend of cells grown in a matrix. This means that organoids contain for instance urothelial cells expressing cytokeratins (CK) [57,59], mesenchymal cells expressing vimentin, endothelial cells or smooth muscle cells expressing phalloidin [57], BC progenitor cells expressing BC stem cell markers such CD24, CD44 [86,87], Ki67^pos^ proliferation competent cells [59], and others. Our study provides evidence that some but not all cells in BCOs express one or several CKs. Antibody AE1/AE3 binds to CK1–8, CK10, CK14–16, and CK19, respectively, and AE1/AE3^pos^ cells were observed. In addition, we detected expression of CK5 and CK20. Basal urothelial cells were shown to express CK5 and/or CK6, while luminal urothelial cells expressed CK20 [88]. Our organoids therefore contain both populations, basal as well as luminal urothelial cells. A stringent distribution of CK5^pos^ versus CK20^pos^ urothelial cells in BCOs was not observed. These distinct cells seem to be distributed randomly. Due to the study design, we cannot determine yet if the basal versus luminal phenotype of the cells in organoids is conserved ex vivo or if it is induced randomly by in vitro cell culture conditions. In addition to the basal and luminal urothelial cell phenotypes, recent studies even suggested additional subtypes of urothelial cells in BC [89,90]. This diversity in urothelial cells is corroborated by our data. Thus, BCOs reflect at least in part the variety of urothelial cells described in situ.

Vimentin expressing mesenchymal cells were detected in BCOs as well. However, expression of vimentin seemed lower in the BCOs investigated when compared to other proteins as extended exposure was needed to document the low fluorescence intensities by microscopy. In contrast to CK^pos^ urothelial cells, much less is known about vimentin^pos^ mesenchymal cells in BCOs. This, in part, may be associated with the composition of the expansion media used for BCO generation. Our very preliminary data suggest that growth of BCOs seems not to be enhanced by addition of a medium optimized for expansion of mesenchymal cells (data not shown). Therefore, vimentin^pos^ cells may be diluted out or overgrown by urothelial cells upon continued organoid passages. However, this must be investigated in more detail in future experiments. In urothelial carcinoma, vimentin was detected only in the invasive front of some of the tumor samples and consequently was not considered a general BC marker [91]. BC tissue samples seem therefore not be enriched for mesenchymal cells. This predication is in line with our observations. However, in mamma carcinoma, vimentin was associated with epithelial-to-mesenchymal transition (EMT), a migratory phenotype, increased risk for metastasis, and therefore important for cancer progression [92]. Detection of vimentin^pos^ cells in BCOs is important in another context as well. Tumor associated fibroblasts are known to modulate the efficacy of anti-cancer drugs [93,94]. When BCOs are employed in drug screens, the mesenchymal cells may grant a better match of this tumor model to the clinical situation when compared to standard cultures of tumor cells. However, due to the design of this study, a reliable conclusion on the influence of mesenchymal cells in BCOs on drug sensitivities cannot yet be drawn.

Bladder cancer stem cells have been described by expression of stem cell markers, among them CD24, CD44, and CD47, respectively [86,95]. BC stem cells were detected in muscle invasive BC samples [96,97]. In some studies, organoids were produced from tissue samples of muscle invasive BC [57,59]. Others used non-invasive papillary urothelial carcinoma and generated spheroids [98]. Of note, this study provides evidence that organoids can be generated from UTUC tissue samples as well. In contrast to BCOs, organoids from upper urinary tract samples are not well studied yet. Moreover, in the 2D as well 3D organoid assays, the two UTUC-derived organoids seemed less sensitive towards CIS, VTX, and S63 when compared to the cells derived from BC samples. The trend observed here can also be associated with the distinct sensitivity of cells from the individual donors. Larger cohorts must be explored to come to conclusive results. To the best of our knowledge, our data provide experimental evidence for the first time that organoids can be generated from UTUC samples employing the methods developed for production of BCOs [57,59]. In addition, such UTUC-derived organoids may serve as tools for therapy testing as well [52]. When following a recent definition to discriminate organoids from spheroids or cell clusters, organoids contain differentiation-competent progenitor cells, form tissue-like structures containing differentiated cells, and produce an extracellular matrix [99]. Upon seeding in Matrigel, a sorted single cell was reported to give rise to an organoid containing different cells and resembling the tissue of origin [100]. Bona fide BC progenitor cells were found recently in BCOs [87]. However, to the best of our knowledge, it has not been investigated in depth if tumor-derived organoids contain a cancer stem cell niche or a functional cancer stem cell niche equivalent. Therapeutic regimens aiming at cancer stem cells should therefore not utilize organoids after extended expansion. In late-stage organoids, the stem or progenitor population may be reduced [87].

In this context, another aspect of organoid biology merits attention as well. Engagement of cells by integrins or other receptors including CD44 influences cell survival, apoptosis, and anoikis [40,101,102]. Thus, integrin signaling may modulate the action of cytotoxic agents [103,104,105,106]. In organoids, cells bind to the extra cellular matrix of the scaffold by integrins and other receptors. Moreover, cells in organoids may complement the Matrigel scaffold by expression of collagens and other matrix components as well. The efficacy of apoptosis-inducing cancer drugs, including for instance cisplatin, may be reduced in this 3D tissue-like tumor model in vitro [107]. However, drug tests with cells in monolayer or in suspension cultures will fail to appreciate this component of matrix-modulated viability signals mediated for instance by the p125 focal adhesion kinase [108]. The differences observed here between cytotoxic effects to the same cells in 2D versus 3D may be in part explained by this difference in intracellular signaling. However, additional research with BCOs is required to make robust conclusions in this context.

By any means, cancer tissue-derived organoids grant complementary in vitro studies of cancer therapy targeting tumor cells in a micro-environment related to the situation of cells in vivo. However, with respect to urothelial cancer-derived organoids, several technical challenges towards this goal remain. In contrast, for instance, to the generation of organoids from breast cancer or gastrointestinal tumor samples, efficacy of the generation of organoids from BC samples seems not yet sufficient [51,56,109]. Moreover, data on UTUC-derived organoids are currently scarce. The protocols reporting on the generation of BCOs yielded quite distinct outcomes [50,57,58,98]. Therefore, differences arising from distinct procedures in the production of BCO in different laboratories cannot be studied in comparison to a standard BCO. Our preliminary data suggest that at least 3D spheroids can be generated, e.g., from urothelial carcinoma cell lines such as HT1197, T24, UC14, UC15, and RT4, and RT112. Such spheroids may serve as surrogate BCO standards to bridge differences observed between studies employing BCOs generated by different protocols [57,59].

## 4. Materials and Methods

We generated organoids from urothelial tumor cells of 7 donors and selected organoids from 3 donors which granted stable growth as organoids and as adherent 2D cultures. We confirmed expression of urothelial antigens and tumor markers by immunohistochemistry and investigated the dose response curves of two novel components, venetoclax (VTX; Selleck Chemicals, Huston, TX, USA) versus S63845 (S63; Selleck Chemicals, Huston, TX, USA), in comparison to a clinical standard, cisplatin (CIS; Sigma-Aldrich Chemie, Taufkirchen, Germany), in organoids in comparison to the corresponding 2D cultures by WST and chemiluminescence viability assays. Normal urothelial cells (NUCs) and tumor lines RT4 and HT1197 (ATCC; Manassas, VA, USA) served as controls. An overview of the study design is presented in Figure 7.

### 4.1. Organoid Culture

Pathology-confirmed bladder cancer tissue samples were obtained from the Dept. of Urology after written and confirmed consent and shipped on wet ice to the laboratory. Organoids were prepared following published protocols [57]. In brief, the tissue was placed in a petri dish to determine the wet weight, covered with PBS, and dissociated by aid of scalpels in tiny pieces. The samples were collected and sedimented by centrifugation (480× *g*, 10 min, ambient temperature). The supernatant was removed, and the pieces were resuspended in HCM (Corning, Glendale, AZ, USA) complemented with type II collagenase (2 × 30 min, 3000 U/mL, 37 °C, 5% CO_2_; STEMCELL, Cologne, Germany). To remove debris, the digested tissue was filtered (100-µm and 70-µm mesh size), and the cells were sedimented by centrifugation (150× *g*, 5 min, ambient temperature). The yield of cells was determined by aid of a hematocytometer using trypan blue dye exclusion. Aliquots of Matrigel (Bio-Techne, Nordenstadt, Germany) were prepared on wet ice. A total of 20,000 cells were resuspended in 10 µL HCM and mixed with 30 μL Matrigel on ice. A total of 40 µL of this blend containing cells and Matrigel were dispensed in one well of a 24-well plate. The plate was flipped headlong 180° to generate a hanging drop and incubated at 37 °C, 5% CO_2_ for 15 min in humidified atmosphere to harden the hydrogel. After that, the plate turned 180° again, and 500 µL of the organoid culture medium (HCM enriched by 5% charcoal-stripped FBS, Sigma-Aldrich Chemie, Taufkirchen, Germany, 0.5 µL Y-27632) were added to each well. The Rock inhibitor Y-27632 (10 µM, MedChemExpress, Monmouth Junction, NJ, USA) was added to complement the organoid culture medium during the first 7 days of incubation to avoid apoptosis of the cells. This study was approved by the local Ethics Committee under file number 804/2020/B02.

### 4.2. Immunofluorescence Assay

Immunofluorescence staining was used to characterize the organoids in 8-well chamber slides (Sarstedt, Nuembrecht, Germany). Prior to the staining, the culture medium was carefully aspirated; organoids were washed twice with PBS at ambient temperature; PBS was aspirated; and organoids were fixed (4% formaldehyde in PBS, 30 min, ambient temperature). The organoids were rinsed twice by PBS, and all liquid was removed. Then, 100 µL of the antibody solution per chamber were added and incubated for 1 h at 37 °C, 5% CO_2_ in a humidified chamber (Table 3). After incubation, the primary antibody solution was removed, and the samples were washed three times for 3 min with 250 µL PBS per chamber. The fluorescence-labeled secondary antibody was added in the presence of DAPI (Table 3), and the samples were incubated at room temperature for 1 h in a humidified chamber in the dark. The antibody/DAPI solution was poured off, and the samples were washed with 250 µL PBS per chamber three times for 3 min. The samples were covered with Dako mounting medium and cover glasses, examined by fluorescence microscopy (Axiovert, Zeiss, Oberkochen, Germany), recorded, and evaluated (ZenBlue, Zeiss).

### 4.3. Cytotoxicity Assay

#### 4.3.1. Drug Testing in 2D Cell Culture

Normal urothelial cells (NUCs) were prepared from ureter samples of patients undergoing kidney surgery at the University Hospital (ethics committee approval #804/2020/B02) and expanded as described [110]. Bladder tumor cell-lines RT4 and HT1197 (ATCC Manassas, VA, USA) were expanded as requested from the supplier. Patient-derived urothelial cancer cell cultures were the other source for adherent cells (=2D). The tissues from the surgery were divided into two parts. One part was utilized for the production of organoids; the remaining part was used for preparation of UCs in 2D cultures. The UCs were characterized and expanded as described [110]. In some cases, the amount of tissue obtained for cell isolation was not sufficient to generate organoids and 2D UC cultures. In these cases, organoids were produced in the first place. Upon splitting, an aliquot of cells was set aside and transferred to 2D cell culture (Figure 7). We considered this a feasible way to facilitate drug testing in 3D vs. 2D cultures. To obtain the number of cells required for 2D drug tests (2000 per well), cells were harvested by trypsin-EDTA, washed, counted, diluted, and resuspended in 100 µL expansion medium, and seeded to a well in a flat-bottom 96 well ELISA plate. All 2D drug analyses with cells were performed in quintuplicates. After overnight incubation, the medium was aspirated and replaced by the expansion medium complemented with different concentrations of CIS (1 to 30 μM), VTX (0.64 to 10 μM for CTGs and 1.6 to 25 μM for WSTs), and S63 (as VTX), respectively, in the concentrations as indicated (Figure 3 and Figure 5). Cells incubated in the medium without drugs or the sovent DMSO served as controls. After incubation of 1 to 4 days, the expansion medium containing the drugs was removed, and the cell-viability was tested by the corresponding reagents (WST, Roche, Basle. Switzerland), CellTiterGlo, Promega, Madison, UN, USA), and an apparatus (GloMax, Promega Madison, UN, USA) following the instruction of the manufacturer.

#### 4.3.2. Drug Testing in Organoids

In addition to the cytotoxicity analyses of adherent cells, effects of CIS, VTX, and S63 on cells in BCOs were investigated. Due to the limitations of the detection technique and according to the guidelines of the reagents applied, the diameter of organoids to be tested by the CellTiter-Glo 3D-chemistry and GloMax apparatus (Promega, Madison, UN, USA) must be limited to less than 300 µm. Therefore, the mean size of BCOs was checked under the microscope before testing the cytotoxicity. After that, the BCOs were degraded by proteolysis (dispase II, 1 h, 37 °C, 5% CO_2_). The dispersed samples were sedimented by centrifugation at 150× *g* for 5 min and resuspended in the organoid culture medium. The organoids were counted and diluted to achieve a BCO suspension for the drug testing containing 1000–2000 organoids per milliliter. Next, 100 µL of the organoid suspension, complemented by 5 µL Matrigel, were seeded in each well. Then, 1 µL of the drug solution was added to each well to yield the drug concentrations desired and incubated for 24 h, 48 h, and 72 h. All drug analyses of organoids were performed as triplicates. The cytotoxicity assay was recorded employing the CTG method. BCOs in the organoid culture medium without drugs or containing solvent 1% DMSO served as controls.

### 4.4. Data Processing and Statistics

To process the results obtained by cytotoxicity experiments, original data were exported to spreadsheet (MS Excel 16.61.1, Microsoft, Albuquerque, NM, USA) or statistics programs (GraphPad Prism 8.0, GraphPad Software, La Jolla, CA, USA) and explored. Mean values of data sets (quintuplicates of cytotoxicity assays of cells in 2D cultures/triplicates of cytotoxicity assays of 3D organoids) were calculated and presented in the figures as dose-response kinetics. The normalized viability index (NVI) was computed by the following formula: NVI = $\frac{\left(mean\;test-blank\right)}{\left(mean\;control-blank\right)}\times 100$ in percent (%). Significant differences were computed by a two-way ANOVA. *p*-values’ summary as 0.01~0.05 (*), 0.001~0.01 (**), 0.0001~0.001 (***), or <0.0001 (****) were considered statistically significant and marked in the artwork accordingly.

## 5. Conclusions

We conclude that organoids can be generated from upper tract urothelial cancer and from bladder cancer tissue samples. They maintained inter-individual sensitivities towards cisplatin, venetoclax, and S63845. Organoids exhibited articulately distinct sensitivities towards the three components investigated when compared to the corresponding 2D cultures. The viability assay and test systems employed yielded a bias to the results. Intensive research to understand better the differences in drug sensitivities observed in our study between 2D and 3D cultures of cells from the same tumor is needed to develop more reliable 3D organoid culture technologies for screening anti-cancer drugs in meaningful ways for an individual suffering from bladder cancer.

## Figures and Tables

**Figure 1 ijms-23-06305-f001:**
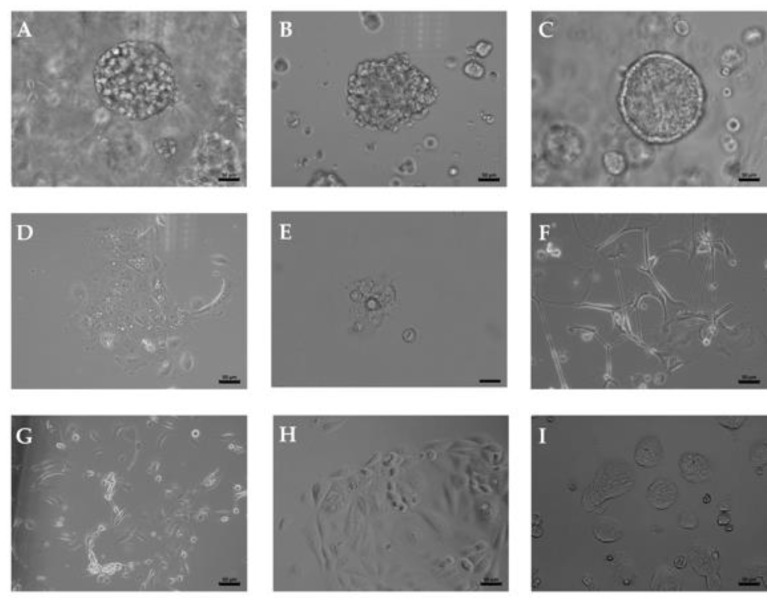
Urothelial carcinoma organoids in culture. Cells from tissue samples derived from BC patient #56 (**A**,**D**), #140 (**B**,**E**), and #147 (**C**,**F**) were expanded as organoids (**A**–**C**) or adherend 2D standard cultures (**D**–**F**). Normal urothelial cells (**G**) and BC cell lines HT1197 (**H**) and RT4 (**I**) served as controls. Size bars indicate 50 μm.

**Figure 2 ijms-23-06305-f002:**
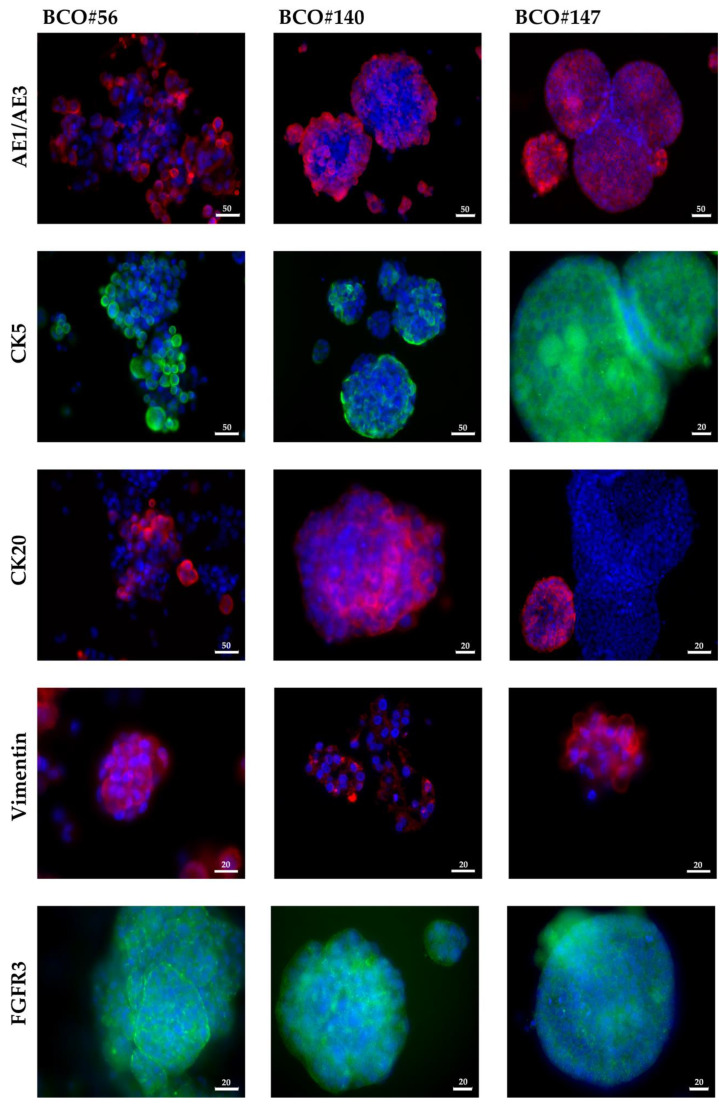
Characterization of urothelial organoids by immunofluorescence. Organoids were fixed and stained by antibodies to epithelial and urothelial lineage markers, i.e., cytokeratins AE1/AE3, CK5, and CD20 as indicated. In addition, expression of the mesenchymal antigen vimentin and of fibroblast growth factor receptor 3 was investigated. Size bars indicate 20 μm or 50 μm as indicated.

**Figure 3 ijms-23-06305-f003:**
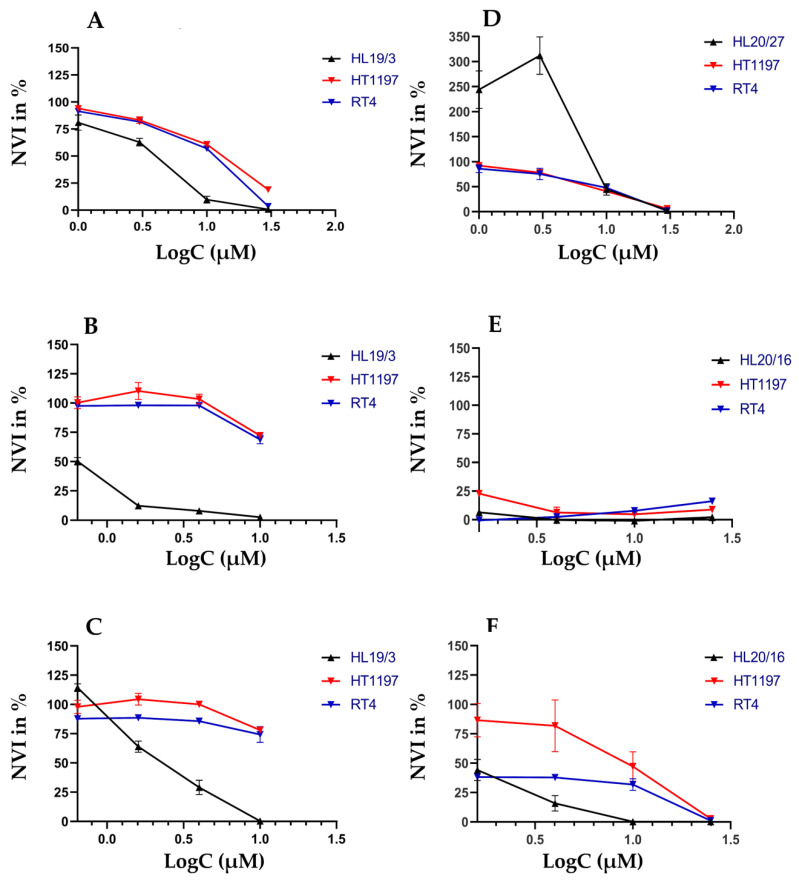
Comparing cytotoxic effects in urothelial cells in 2D cultures by two different assay types. Normal urothelial cells (NUCs HL19/3, HL20/16, HL20/27) as well as bladder cancer cell lines HT1197 and RT4 were incubated with different concentrations of CIS (**A**,**D**), VTX (**B**,**E**), S63 (**C**,**F**) to determine the cell viability after two days of incubation by either a chemiluminescence assay (CTG 2.0; **A**–**C**) or a colorimetric assay (WST; **D**–**F**). The mean normalized viability index in percent (NVI %; ordinate, *Y*-axis) is disclosed as function of the logarithm of drug concentrations employed (LogC; (µM), abscissa, *X*-axis).

**Figure 4 ijms-23-06305-f004:**
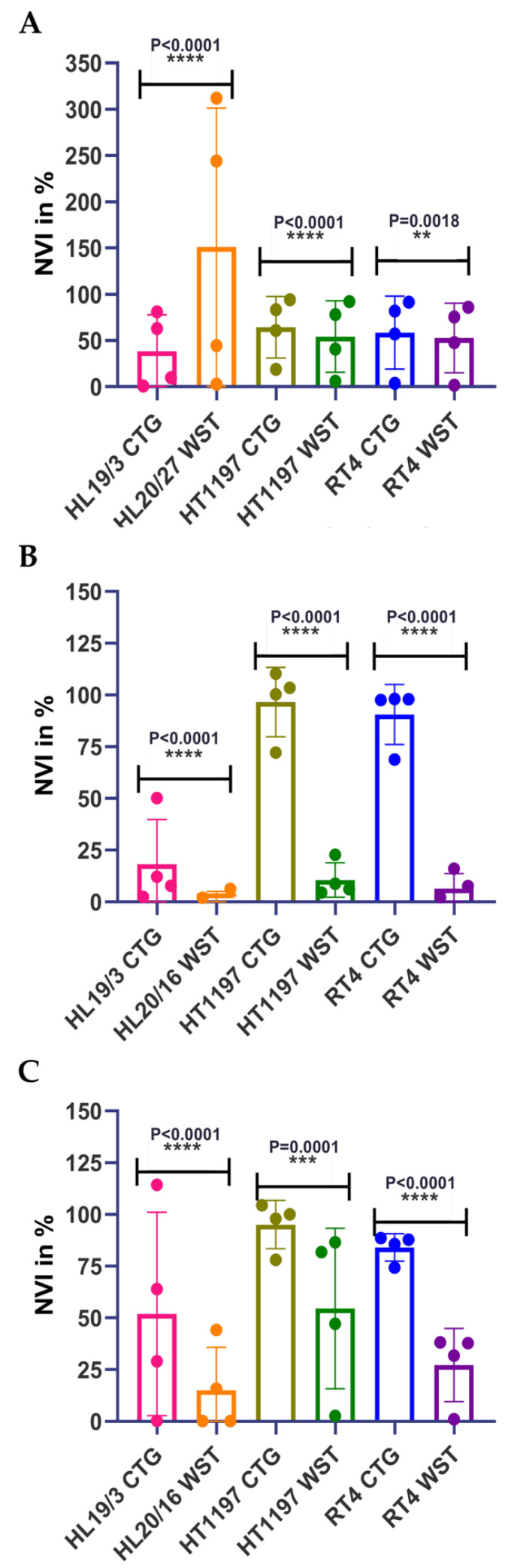
Comparing the normalized cell viabilities upon tests with different assays. NUCs (HL19/3, HL20/16, HL20/27) and BC lines HT1197 and RT4 were incubated with different concentration of CIS (**A**), VTX (**B**), and S63 (**C**), and cell viabilities were compared after incubation for 1–4 days employing the CTG chemiluminescence assay or the WST colorimetric assay. ** *p* = 0.0018, *** *p* = 0.0001, **** *p* < 0.0001.

**Figure 5 ijms-23-06305-f005:**
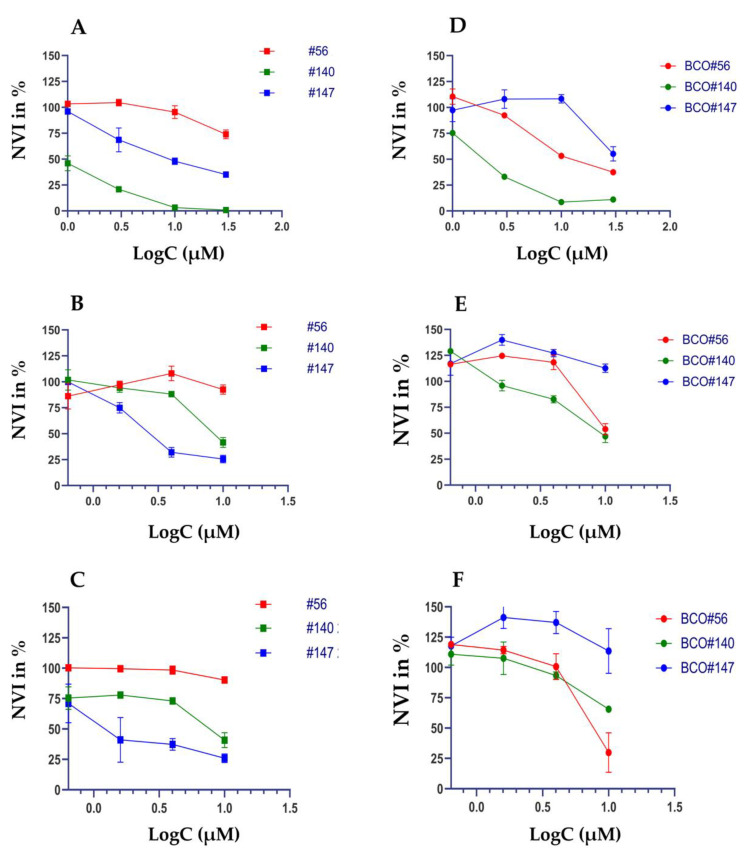
Comparing cytotoxic effects in urothelial carcinoma cells in 2D versus 3D cultures. BCOs were generated from cells of patients #56, #140, and #147 and expanded. From BCOs, cells were derived for further expansion in adherent 2D cultures (**A**–**C**) or 3D organoid cultures (**D**–**F**). The CTG chemiluminescence assay in 2D (left panel) vs. 3D (right panel) cultures was performed to compare the responses of the cells to CIS (**A**,**D**), VTX (**B**,**E**), and S63 (**C**,**F**), respectively. The mean normalized viability index in percent (NVI %; ordinate, *Y*-axis) is disclosed as function of the logarithm of drug concentrations employed (LogC; (µM), abscissa, *X*-axis).

**Figure 6 ijms-23-06305-f006:**
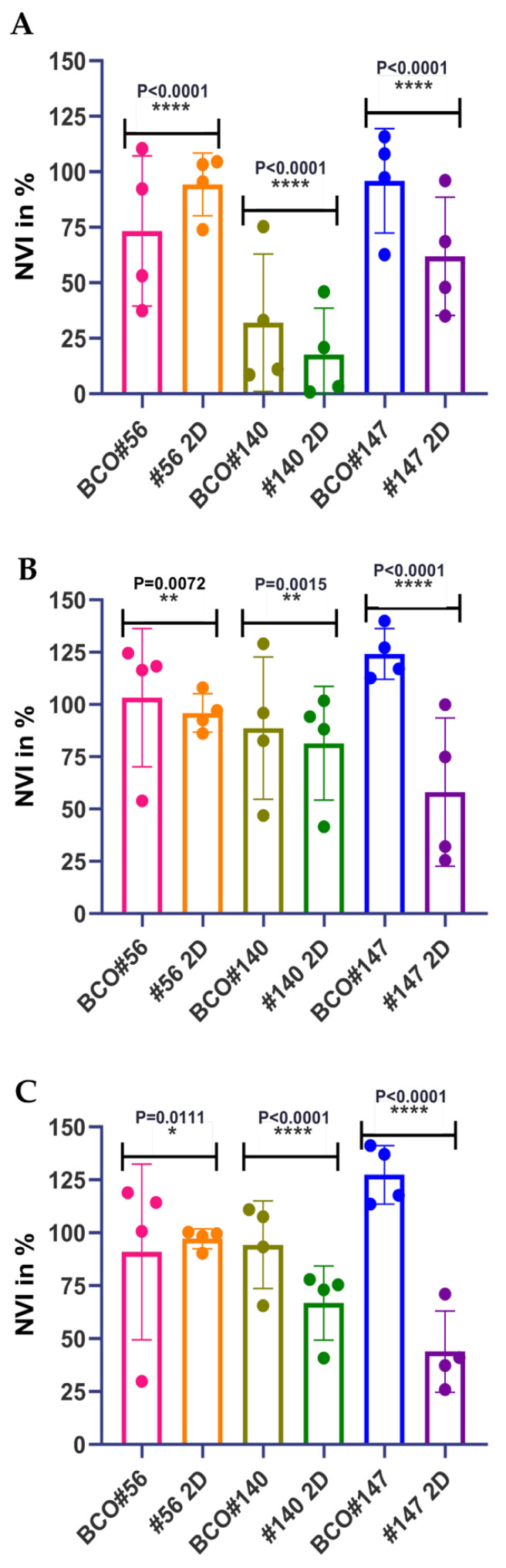
Comparing cell viabilities after drug treatment in organoids and adherent cells. BCOs and the corresponding tumor cells were incubated for 1, 2, and 3 days with CIS (**A**), VTX (**B**), and S63 (**C**), respectively, to determine the cell viability in 3D organoids in comparison to the same cells in adherent 2D standard culture. The mean normalized viability indices in percent (NVI %; ordinate, *Y*-axis), statistical differences, and *p*-values were computed in the corresponding cohorts as indicated (abscissa, *X*-axis). * *p* = 0.0111, ** *p* = 0.0072 & 0.0015, **** *p* < 0.0001.

**Figure 7 ijms-23-06305-f007:**
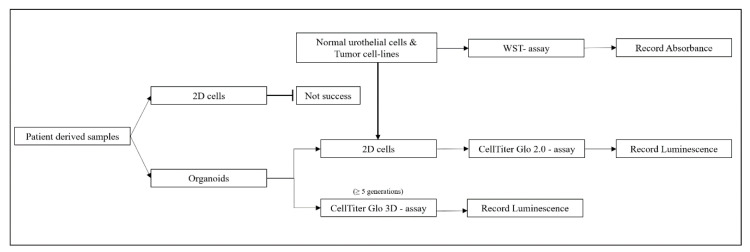
Graphical overview over the study design.

**Table 1 ijms-23-06305-t001:** Half maximal inhibitory concentrations of CIS, VTX, and S63 in urothelial cells in adherent culture explored by two different viability assays ^1^.

IC_50_ (µM) Cells, Time	Cisplatin WST/CTG	Venetoclax WST/CTG	S63845 WST/CTG
NUCs, day1	8.41/19.01	1.18/6.21	~4.15/3.29
NUCs, day2	~9.71/3.67	0.21/0.74	2.37/2.12
NUCs, day3	~9.83/3.52	6.1 × 10^152^/0.63	2.2 × 10^151^/2.41
HT1197, day1	22.38/91.56	2.38/~252,423	16.26/~12.05
HT1197, day2	10.50/17.76	4.6 × 10^150^/~12.16	5.48/~11.51
HT1197, day3	7.18/12.05	0.38/~10.71	8.62/~11.49
RT4, day1	18.85/20.45	41.88/~11.76	10.81/755.70
RT4, day2	10.06/11.82	85.27/13.70	7.24/59.68
RT4, day3	7.27/9.96	113.60/12.78	5 × 10^143^/195.00

^1^ Mean of half maximal concentrations (IC_50_, µM) of the 3 drugs determined by quintuplicate assays of a representative analysis. WST: colorimetric assay with water-soluble tetrazolium. CTG: CellTiterGlo, chemi-luminescence assay.

**Table 2 ijms-23-06305-t002:** Comparing drug effects in 2D versus 3D cultures.

Cells, Culture	Cisplatin	Venetoclax	S63845
Cells #56, 2D	49.82	0.32	29.77
BCO #56, 3D	15.48	10.08	9.30
Cells #140, 2D	0.89	8.73	8.88
BCO #140, 3D	2.00	9.27	12.79
Cells #147, 2D	10.85	3.15	1.63
BCO #147, 3D	~30.35	~2.9 × 10^0.48^	~

Growth inhibition as half maximal inhibitory concentrations (IC_50_) of the corresponding drug in μM towards cells from the same patient in 2D vs. 3D.

**Table 3 ijms-23-06305-t003:** Reagents for immunofluorescence staining.

Antibody	Supplier
Primary antibodies:	
Mouse-anti-CK antibody AE1/AE3 (MAB3412)	Millipore, Taufkirchen, Germany
Rabbit-anti-CK5 (905504)	BioLegend, San diego, CA, USA
Mouse anti-CK8 antibody (MA5-14088)	Invitrogen, Waltham, MA, USA
Mouse anti-CK20 (M7019)	Dako, Jena, Germany
Mouse Anti-Vimentin (550513),	Becton Dickinson, Heidelberg, Germany
Rabbit anti-FGFR3	Invitrogen, Waltham, MA, USA
Secondary antibodies:	
Goat-anti-mouse IgG Cy3	Jackson ImmunoResearch, Cambridgeshire, UK
Goat-anti-rabbit IgG Alexa FI.488	Jackson ImmunoResearch,
DAPI	Jackson ImmunoResearch

Primary antibody AE1/AE3, antibodies to CK5, CK20, FGFR3, and Vimentin, as well as the isotype control were diluted 1:100 in 1% BSA/PBS-T. DAPI, Alexa- and Cy3-fluorescence-labeled secondary or detection antibodies were diluted 1:1000 in 1% BSA/PBS-T.

## Data Availability

Original data will be disclosed to colleagues in academic and public research institutions upon comprehensible request for scientific purposes only.

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
