# Peer review of "Urinary Tract Tumor Organoids Reveal Eminent Differences in Drug Sensitivities When Compared to 2-Dimensional Culture Systems"

_ijms, 2022, doi:10.3390/ijms23116305_

Round 1
Reviewer 1 Report
This is an interesting and well written article, with potential impact on how cancer drugs are tested in vitro and on the development of more targeted treatments for urinary cancers.
I have the following comments, mainly relating to Figure 3, that need addressing before publication:
- line 39. 'among the most frequent malignancies recorded' it is a bit of an overstatement as they are relatively uncommon compared with cancers such as breast, prostate and lung
- Figure 3. How many replicate experiments were performed? Was statistical analysis performed? Were the differences between cell lines/treatments statistically significant?
- Figure 3. How is the NVI in% calculated? Should the NVI be approximately 100% at zero, indicating that cells were healthy and viable before treatment? In some cases the value at zero seems to be very low, especially in panel E, suggesting that the viability of the cells was already affected before treatment. In some cases the value at zero was different for different cell lines e.g. panel B and F. Can different cell lines be compared when the NVI % differ even before treatment?
- Figure 3. Why was a different NUC line used in the two assays (A-B-C vs D-E-F)?
- As organoids are composed by different cell types, can the effect of treatment be influenced by the proportion of cell types/populations? In which case, do the proportion truly reflect those in the patient tumour and therefore would the effect seen in the organoids really reflect effectiveness of treatment in the patient? This should be commented on in the discussion.
- Only one type of cytotoxic assay (CTG assay) was used with the organoids, therefore there is no evidence on whether the results are reproducible with other methods or vary as it was observed in the case of 2D cultures when different methods were applied (Figure 2). This should be commented on, as it is a limitation of the study.
- line 286. Spelling mistake ? - 'mama' carcinoma
Author Response
Reviewer 1:
This is an interesting and well written article, with potential impact on how cancer drugs are tested in vitro and on the development of more targeted treatments for urinary cancers.
I have the following comments, mainly relating to Figure 3, that need addressing before publication:
1.) line 39. 'among the most frequent malignancies recorded' it is a bit of an overstatement as they are relatively uncommon compared with cancers such as breast, prostate and lung
ad 1: We appreciate this statement as it corrects a somewhat misleading statement. In lines 26/27 of the revised manuscript the new wording is found.
2.) Figure 3. How many replicate experiments were performed? Was statistical analysis performed? Were the differences between cell lines/treatments statistically significant?
Ad 2: We thank reviewer 1 for this valuable suggestion. To our own detriment, the individual number of tests performed had been disclosed only in part in the footnote to table 1 of the original study. To provide the information necessary on how many replicates were employed in the drug tests, we complemented the revision in the section M&M in lines 405/406, 426/42, and 432/433, respectively. Moreover, a new chapter was added during revision to describe data processing and statistics in more detail (see M&M, section 4.4. lines 430 to 436).
3.) Figure 3. How is the NVI in% calculated? Should the NVI be approximately 100% at zero, indicating that cells were healthy and viable before treatment? In some cases the value at zero seems to be very low, especially in panel E, suggesting that the viability of the cells was already affected before treatment. In some cases the value at zero was different for different cell lines e.g. panel B and F. Can different cell lines be compared when the NVI % differ even before treatment?
Ad 3: This information is added to the new section 4.4 of the revised version of our manuscript, including the formula: NVI%= . Moreover, the values presented in the X-axis in figure 3 are presented in a Log scale. Therefore, at the zero point presented “1µM”, log(1)=0. The natural value “0” (zero) is not defined in log scales. Therefore, viability counts w/o drugs were not included in the calculation and graph. In the Y-axis cell viability values ( = NVI) will start at already low values, if the drug is very efficient. In contrast, cisplatin is less potent and therefore an NVI of 100% is seen in our experiment, as for instance presented in figure 3 A – cisplatin treatment - and CTG chemistry. Here we approach to “100%” viability upon normalization of values of treated cells to control group based on the mathematical calculations.
4.) Figure 3. Why was a different NUC line used in the two assays (A-B-C vs D-E-F)?
Ad 4: We performed different parts of the experiments depending on the availability of BOCs and NUCs in different time spans. In addition, several preliminary series of experiments had to be performed to establish the assays needed for our study. For instance, seeding densities of NUCs in 96-well plates for sufficient read-out by WST and GloMax after 1 – 4 days of incubation ± drugs had to be tested prior to setting up the actual experiments intended. In contrast to the tumor lines, NUCs undergo cellular senescence upon extended expansion. To avoid artifacts associated with retarded spontaneous proliferation of senescent cells, we preferred to employ NUCs in comparably early passages from different donors over NUCs from 1 donor after different population doublings. The results section was revised to address this point in lines 146 – 148.
5.) As organoids are composed by different cell types, can the effect of treatment be influenced by the proportion of cell types/populations? In which case, do the proportion truly reflect those in the patient tumour and therefore would the effect seen in the organoids really reflect effectiveness of treatment in the patient? This should be commented on in the discussion.
Ad 5: Staining of organoids with mesenchymal and urothelial markers clearly supports the thought of reviewer 1: BCOs contain different types of cells (compare Fig. 2 of the manuscript) and therefore may reflect the tissue situation of bladder cancer much better than standard 2 D cultures of cancer cells that may be enriched by fast growing and firmly attaching cells. However, as of today, our data do not allow for analyses along the lines suggested by reviewer 1. But this reviewer raised a very interesting question that needs additional analyses. We are thankful for this comment and hope to tackle it by our current and future experiments.
6.) Only one type of cytotoxic assay (CTG assay) was used with the organoids, therefore there is no evidence on whether the results are reproducible with other methods or vary as it was observed in the case of 2D cultures when different methods were applied (Figure 2). This should be commented on, as it is a limitation of the study.
Ad 6: This is a very important point the reviewer raised. Different commercial providers offer other technologies than CTG (Promega) to evaluate cell viabilities in organoids as well. Regrettably we cannot use another chemistry in our devices, nor other apparatus incl. specific reagents for additional cytotoxicity assays. We therefore cannot compare our cytotoxicity data generated by CTG apparatus and chemistry with the outcome using other systems. Therefore, a comment was added to the discussion (lines 234 – 237) to properly address this important point.
7.) line 286. Spelling mistake ? - 'mama' carcinoma
ad 7: excellent catch. Thank you. The missing “m” was added to the revision.
Reviewer 2 Report
The manuscript describes the usefulness of three-dimensional tumor organoids derived from human urothelial tumors to test their sensitivity against three different drugs. The results were compared to those obtained using the same cells in two-dimensional cultures. Cultures from tumor-derived patients were also compared to tumor cells lines. The objective of the work is clear. As the authors indicated this model could be useful as a previous screening step before assaying them in in vivo assays. The methods are simple and well explained and performed to demonstrate the and correctly performed to demonstrate the usefulness of organoids cultures. However, the number of replicas and the statistical differences among the results obtained are not indicated, invalidating the conclusions. Some explanations are also needed to clarify some of the methods and results reported.
Abstract
- It can be improved. For instance, lines 27-28 and lines 33-34 repeat the same results.
- There are some abbreviations that should be spelt out before using.
Introduction
- Lines 67 The schedules of the treatments are not clear. There is confusing the explanation of BC treatment. I recommend emphasize in which cases for instance cisplatin is used, because is one of the drugs that is used in the present study. and the of treatment should be updated. In lines
- Explain the relevance of cisplatin in non-muscle invasive bladder cancer too, since in the experiments the RT4 cell line is used. RT4 are grade 1 tumor cells characteristics from a non-invasive phenotype.
- Unify the nomenclature of the drugs used in the study. VTX and S63 are named in the introduction section, but the name is different in the abstract section. Unify along through the manuscript. The full name of the drugs is in the material and methods section that it is at the end of the manuscript.
- In the introduction section, change RT112 by RT4 cell line, that it is used as a control cell line. RT112 do not appear in the rest of the manuscript.
Results
- Figure 1 and 2. Are the images representative from a series of experiments? Are the images taken from an unique culture? Indicate this.
- Table 1. Re-name the title of the table indicating exactly which data is being compared.
- Table 1 and 2. Indicate in a table legend the meaning of the abbreviations.
- Table 2. Indicate which values are indicated in this table. Percentage of growth inhibition?
- Indicate in Figures 3 and Figure 4, which is the meaning of some abbreviations, for instance indicate what is the measure in y axis.
- In the Figure 4 the figure legend of the left panel is not complete.
- The results obtained are only based on the observation of the cytotoxic curves obtained. There is none statistical analysis of the results. To confirm the conclusion drawn from the results obtained, there is mandatory to perform a statistical analysis and to observe significant differences between both assays (2D and 3D)
Material and methods
- To validate the results, the number of replicates and/or experiments performed in each case should be indicated. It is only mentioned as a note in Table 1.
Author Response
The manuscript describes the usefulness of three-dimensional tumor organoids derived from human urothelial tumors to test their sensitivity against three different drugs. The results were compared to those obtained using the same cells in two-dimensional cultures. Cultures from tumor-derived patients were also compared to tumor cells lines. The objective of the work is clear. As the authors indicated this model could be useful as a previous screening step before assaying them in in vivo assays. The methods are simple and well explained and performed to demonstrate the and correctly performed to demonstrate the usefulness of organoids cultures. However, the number of replicas and the statistical differences among the results obtained are not indicated, invalidating the conclusions. Some explanations are also needed to clarify some of the methods and results reported.
Abstract
- It can be improved. For instance, lines 27-28 and lines 33-34 repeat the same results.
We thank the reviewer for this suggestion and revised the abstract accordingly (see lines 32 – 34 of the revised manuscript).
- There are some abbreviations that should be spelt out before using.
We totally agree to this point of critique and revised the whole text accordingly.
Introduction
- Lines 67 The schedules of the treatments are not clear. There is confusing the explanation of BC treatment. I recommend emphasize in which cases for instance cisplatin is used, because is one of the drugs that is used in the present study. and the of treatment should be updated.
We thank the reviewer for this suggestion and revised the manuscript accordingly to stress the role of cisplatin in the perioperative setting and outline the future role of an individualized approach (see lines 73 -81). In addition, 2 new references (#26 #27) were added to reflect these changes.
- Explain the relevance of cisplatin in non-muscle invasive bladder cancer too, since in the experiments the RT4 cell line is used. RT4 are grade 1 tumor cells characteristics from a non-invasive phenotype.
We thank the reviewer for this comment. The use of RT4 as one of the cell lines was based on the known relative resistance of the cell line in previous studies. We stressed this reasoning in line 147– 148 and added additional references (# 68, #69).
- Unify the nomenclature of the drugs used in the study. VTX and S63 are named in the introduction section, but the name is different in the abstract section. Unify along through the manuscript. The full name of the drugs is in the material and methods section that it is at the end of the manuscript.
Clear and distinct nomenclature is of central importance in science. We agree to this suggestion. We revised the manuscript based on this suggestion of reviewer 2. In addition, we have corrected the differences in nomenclature of VTX and S63 throughout the manuscript, and hope that we have not introduced novel mistakes by these revisions.
- In the introduction section, change RT112 by RT4 cell line, that it is used as a control cell line. RT112 do not appear in the rest of the manuscript.
We apologize for this typographical error and corrected it in the revision R1.
Results
- Figure 1 and 2. Are the images representative from a series of experiments? Are the images taken from an unique culture? Indicate this.
The micrographs in Figure 1 are representative for the cells included. They were taken randomly only to demonstrate that the cells expandable as BCOs were also growing in adherent culture prior to initiating the cytotoxicity assays in 2D versus 3D. They by no means were meant to explore differences in composition of the populations, their phenotype, metabolism etc. in great details.
The micrographs shown in figure 2 are representative pictures as well. But they were taken from organoids specifically seeded in 8-chamber chamber slides for staining purposes. During the whole study, the BCOs were seeded in chamber slides and utilized for immunofluorescence analyses several times. They therefore are not the result of a single -one times only – experiment. The results section was revised accordingly (see lines 142 and 147/148, respectively).
- Table 1. Re-name the title of the table indicating exactly which data is being compared
We completely agree to this point. Originally, we were shy to explain in the title of table 1 the details of the experiments and therefore preferred a short title pointing towards “cell viability” and added more information to the corresponding footnote. But it is for sure better to revise the title of table 1 to correctly describe the data presented. We thank reviewer 2 for this suggestion.
- Table 1 and 2. Indicate in a table legend the meaning of the abbreviations.
To the best of our knowledge, the IJMS prefers not to add legend to tables. The journal prefers footnotes. We therefore extended the footnotes during this revision to pack some of the information suggested in the footnotes of tables 1 and 2.
- Table 2. Indicate which values are indicated in this table. Percentage of growth inhibition?
Excellent catch! We thank reviewer 2 for this detailed critique as we have obviously overseen an important point. We revised both, title, and footnotes of table 2 to better present the data listed in this table.
- Indicate in Figures 3 and Figure 4, which is the meaning of some abbreviations, for instance indicate what is the measure in y axis.
Again, a very good suggestion to really improve the quality of our manuscript. The legends were revised accordingly.
- In the Figure 4 the figure legend of the left panel is not complete.
We did not quite get this point of critique, but we guessed that the wording in the third sentence of legend 4 was confusing. We revised the legends of figure 4 to the best of our understanding and hope that we included by our revision the flaw reviewer 2 referred to.
- The results obtained are only based on the observation of the cytotoxic curves obtained. There is none statistical analysis of the results. To confirm the conclusion drawn from the results obtained, there is mandatory to perform a statistical analysis and to observe significant differences between both assays (2D and 3D)
We do not quite agree to this point and therefore added two new figures to the revision R1. In the new figures we present detailed statistics on drug effects in 2D vs. 3D analyses. They are presented in the revised version R1 in the results section (see Fig. 4new: p. 8/28 and lines 225 – 231 as well as Fig. 6new: p11/28 and lines 301 -311).
Material and methods
- To validate the results, the number of replicates and/or experiments performed in each case should be indicated. It is only mentioned as a note in Table 1.
We have addressed this point of critique and complemented M&M accordingly page 17/28; lines 600/601 and 623/624 In addition, a new section 4.4. was added describing the data computing and statistics was added on pages 17 and 18 of revision R1 (lines 628 – 638).
Round 2
Reviewer 2 Report
The manuscript has been substantially improved after adressing the revierwer's comments.